# Mean Field Models for Neural Networks in Teacher-student Setting

## Abstract

Mean field models have provided a convenient framework for understanding the training dynamics for certain neural networks in the infinite width limit. The resulting mean field equation characterizes the evolution of the time-dependent empirical distribution of the network parameters. Following this line of work, this paper first focuses on the teacher-student setting. For the two-layer networks, we derive the necessary condition of the stationary distributions of the mean field equation and explain an empirical phenomenon concerning training speed differences using the Wasserstein flow description. Second, we apply this approach to two extended ResNet models and characterize the necessary condition of stationary distributions in the teacher-student setting.

## 1 Introduction

In the past several years, neural networks have achieved tremendous successes in many areas of machine learning and artificial intelligence (Devlin et al., 2018; He et al., 2016; Silver et al., 2016; Vaswani et al., 2017), based on remarkable developments in algorithms (Robbins & Monro, 1951; Duchi et al., 2011; Rumelhart et al., 1988), software (Paszke et al., 2017; Jia et al., 2014; Abadi et al., 2016), and hardware (Lindholm et al., 2008; Jouppi et al., 2018). In spite of the empirical successes, the theoretical understanding of neural networks is rather incomplete. To bridge this gap, tools from different fields, such as statistical mechanics, partial differential equations, dynamics systems, have been brought into use. Recently, a series of remarkable papers (Chizat & Bach, 2018; Mei et al., 2018; Rotskoff & Vanden-Eijnden, 2018; Sirignano & Spiliopoulos, 2018) analyze the two-layer neural networks using mean field models and nonlinear partial differential equations (PDEs). The main result is that, under the large width limit and sufficient small learning rates, stochastic gradient descent (SGD) dynamics of a two-layer fully connected neural network with a convex loss can be modeled as a gradient flow of a convex functional of probability distributions under the Wasserstein metric (Peyré et al., 2019; Santambrogio, 2015; Villani, 2008). Though only for two-layer networks, this result significantly simplifies our understanding of the training dynamics. In two recent papers (Nguyen, 2019; Sirignano & Spiliopoulos, 2019), this approach has been extended to multi-layer fully-connected networks by taking the widths to infinity successively from higher to lower layers.

**Contributions.** This paper follows this line of research, focusing on the teacher-student setting. The teacher network is typically a narrow network that computes the function to be learned, while the student network is a vastly over-parameterized wide network. The goal is to optimize the parameters of the student in order to produce the function represented by the teacher. Though somewhat restricted, the teacher-student setting allows for assessing performance by comparing the parameters of the networks. The main contributions of this paper are listed as follows.

First, for the two-layer networks in the teacher-student setting, we characterize for the first time the necessary condition for a stationary distribution of the mean field PDE and show how it relates to the parameters of the teacher network. We also discuss how different activation functions influence the results. Using the Wasserstein flow description, we also provide a simple explanation for why the convergence to teacher parameters with larger weight is much faster than to the ones with smaller weight.

Second, we extend the mean field analysis to two extended ResNet models: the ensemble-ResNet (an average of a large number of independent ResNets) and the wide-ResNet (a ResNet with wide residual blocks). We derive mean field PDEs for the training dynamics of the parameters in both

cases and also characterize the necessary condition for a stationary distribution for the ensemble-ResNet model as well as the first and second order approximations of the wide-ResNet model. Though our derivation remains formal, this is the first time that the mean field model of the ResNet has been considered.

**Related work.** Besides the aforementioned papers on the mean field analysis for neural networks, tools from stochastic analysis (Mandt et al., 2015; Li et al., 2017; 2019; Yaida, 2018; An et al., 2018) and Wasserstein geometry (Arjovsky et al., 2017; Cuturi, 2013; Frogner et al., 2019; Peyré et al., 2019; Solomon et al., 2014) have been playing an increasingly prominent role in machine learning and artificial intelligence.

Another direction that considers the infinite width neural networks is the work (Jacot et al., 2018) of the neural tangent kernel (NTK), which describes under an appropriate scaling the evolution of the function represented by the neural network. At the infinite width limit, the NTK converges to a deterministic limit that depends on the nonlinear activation function and remains unchanged during the training process. Though it is not yet clear how this limit relates to the actual training dynamics of neural networks used in practice (Chizat et al., 2019), mathematically it is an interesting regime that has inspired a lot of recent work (Arora et al., 2019a;b; Cao & Gu, 2019).

In recent years, particle-based methods have also been widely applied to many machine learning problems, such as in Bayesian inference. One such recent example is the Stein variational gradient descent (SVGD) method (Liu & Wang, 2016; Liu, 2017) that approximates a target distribution with a group of interacting particles using iterative gradient updates. It has been shown that, in the large particle number limit, the empirical distribution of the parameters can be modeled by a nonlinear PDE, which can be viewed as a gradient flow of a convex loss function under a Wasserstein-type but non-local metric (Liu, 2017; Lu et al., 2019).

**Organization.** The rest of the paper is organized as follows. Section 2 concerns the mean field description of two-layer networks and provides necessary conditions for stationary distributions. Section 3 extends the analysis to two extended ResNet models. Finally, Section 4 discusses some future directions.

## 2 TWO-LAYER NETWORK IN TEACHER-STUDENT SETTING

### 2.1 TWO-LAYER NETWORKS

**Setup.** Given a function $f : \mathbb{R}^d \to \mathbb{R}$ of the independent variable $x \in \mathbb{R}^d$, consider its approximation by a two-layer network model $\frac{1}{N} \sum_{j=1}^{N} w_j \sigma(x \cdot \theta_j)$, where $N$ is the number of hidden units, $\{\theta_j\}$ are the parameters in a domain $\Theta \subset \mathbb{R}^d$, $\{w_j\}$ are the weights, and $\sigma : \mathbb{R} \to \mathbb{R}$ is an activation function such as rectified linear unit (ReLU), softplus, etc. The following discussion focuses on the special case where the weights $w_j$ are fixed to be one, i.e.,

$$\frac{1}{N} \sum_{j=1}^{N} \sigma(x \cdot \theta_j). \tag{1}$$

The more general case of trainable $w_j$ values are discussed in Appendix 5.1. Assuming that the input $x$ is sampled from a measure $\mu(x)$ independently, the learning problem of this two-layer network is to minimize the loss $E(\theta_1, \ldots, \theta_N) = \int_x \ell \left( \frac{1}{N} \sum_{j=1}^{N} \sigma(x \cdot \theta_j) - y(x) \right) \mu(x) \mathrm{d}x$, where $\ell(\cdot)$ is a loss function. The rest of this paper focuses for example on the quadratic loss case, i.e., $\ell(z) = \frac{1}{2}|z|^2$:

$$E(\theta_1, \ldots, \theta_N) \equiv \frac{1}{2} \int_x \left| \frac{1}{N} \sum_{j=1}^{N} \sigma(x \cdot \theta_j) - y(x) \right|^2 \mu(x) \mathrm{d}x \equiv \frac{1}{2} \left\| \frac{1}{N} \sum_{j=1}^{N} \sigma(x \cdot \theta_j) - y(x) \right\|_{\mu}^2.$$

When optimized with a standard algorithm such as gradient descent (GD), each parameter $\theta_j$ for $1 \leq j \leq N$ is updated with

$$\theta_i[k+1] = \theta_i[k] - \alpha \frac{\partial E}{\partial \theta_i}[k], \quad \frac{\partial E}{\partial \theta_i}[k] = \int_x \left( \frac{1}{N} \sum_{j=1}^{N} \sigma(x \cdot \theta_j[k]) - y(x) \right) \frac{1}{N} \sigma'(x \cdot \theta_i[k]) x \mu(x) \mathrm{d}x,$$

where $\alpha$ is the learning rate of the iterative algorithm. For the SGD, the right hand side is supplemented with a mean-zero noise term. When the learning rate $\alpha$ is sufficiently small, the iterates $\{\theta_i[k]\}$ can be modeled by a continuous-time ordinary differential equation (ODE) system $\{\theta_i(t)\}$ by identifying $t = k\alpha/N$

$$\frac{\mathrm{d}\theta_i(t)}{\mathrm{d}t} = -\int_x \left(\frac{1}{N}\sum_{j=1}^N \sigma(x \cdot \theta_j(t)) - y(x)\right) \sigma'(x \cdot \theta_i(t)) x \mu(x) \mathrm{d}x. \tag{2}$$

**Mean field equation.** Viewing $\theta_i(t)$ as interacting particles and considering the empirical distribution $\rho(t,\theta) \equiv \frac{1}{N}\sum_{j=1}^N \delta_{\theta_j(t)}(\theta)$, we can rewrite $\frac{1}{N}\sum_j \sigma(x \cdot \theta_j) - y(x)$ as $\int \sigma(x \cdot \theta)\rho(\theta)\mathrm{d}\theta$ and $E(\theta_1, \ldots, \theta_N)$ as

$$E(\rho) \equiv \frac{1}{2}\left\| \int \sigma(x \cdot \theta)\rho(\theta)\mathrm{d}\theta - y(x) \right\|_\mu^2.$$

The mean field evolution equation for $\rho(t,\theta)$ associated with (2) is

$$\rho_t(\theta,t) = \mathrm{div}_\theta\left[\rho(\theta,t)\int_x \left(\int_{\theta'} \sigma(x \cdot \theta')\rho(\theta',t)\mathrm{d}\theta' - y(x)\right) \sigma'(x \cdot \theta) x \mu(x)\mathrm{d}x\right].$$

Noticing that the functional derivative of $E(\rho)$ with respect to $\rho$ is given by

$$\frac{\delta E(\rho)}{\delta\rho} = \int_x \left(\int_{\theta'} \sigma(x \cdot \theta')\rho(\theta')\mathrm{d}\theta' - y(x)\right) \sigma(x \cdot \theta)\mu(x)\mathrm{d}x,$$

one can write the mean field PDE for $\rho$ compactly as

$$\rho_t = \mathrm{div}_\theta\left[\rho\nabla_\theta\frac{\delta E}{\delta\rho}\right], \tag{3}$$

which can be interpreted as the gradient flow of the functional $E(\rho)$ under the the Wasserstein metric $(\nabla_\theta^\mathsf{T}\rho\nabla_\theta)^+$ (Jordan et al., 1998; Otto, 2001; Otto & Villani, 2000).

Though the above derivation is mostly formal, one can show that the dynamics of $\theta_i(t)$ converges to the evolution of $\rho(\theta,t)$ for any fixed period of time $[0,T]$ when the number of hidden units $N$ goes to infinity. This is formalized for example in the following theorem.

**Theorem 1** (Reformulated from (Chizat & Bach, 2018)). *If $\theta_i(0)$ are sampled independently from a density $\rho_0(x)$ at time $t = 0$, then for any fixed $T > 0$ the measure $\frac{1}{N}\sum_{j=1}^N \delta_{\theta_j(t)}(\theta)$ converges weakly to the solution $\rho(t,\theta)$ of (3) for any $t \in [0,T]$ as $N$ goes to infinity.*

For the case of bounded parameter domain $\Theta$, such as a ball in $\mathbb{R}^d$ with a fixed radius, one needs to specify the appropriate boundary conditions both for the ODE system (2) and the mean field PDE (3). First, the dynamics of $\theta_i(t)$ should follow normal reflection when $\theta_i(t)$ hits the boundary $\partial\Theta$ of $\Theta$. Second, for the mean field equation of $\rho(t,\theta)$, the Neumann boundary condition $\frac{\partial\rho(t,\theta)}{\partial n}(t,\theta) = 0$ should be used at $\theta \in \partial\Theta$. The bounded case is particularly relevant to the discussion in Section 3.2.

**Stationary distribution in teacher-student setting.** Given the PDE formulation of the parameter evolution, one natural question is to characterize the stationary distribution of (3). In what follows, we address this question in the teacher-student setting, i.e., there exists a parameter distribution $\rho_*(\theta)$ such that the target function $y(x)$ satisfies

$$y(x) = \int_\theta \sigma(x \cdot \theta)\rho_*(\theta)\mathrm{d}\theta.$$

With this teacher representation, the loss $E(\rho)$ can be written equivalently as

$$E[\rho] = \frac{1}{2}\int_x \left(\int_\theta \sigma(x \cdot \theta)(\rho - \rho_*)(\theta)\mathrm{d}\theta\right)^2 \mu(x)\mathrm{d}x.$$

The following theorem characterizes the stationary distribution in this teacher-student setting.

**Theorem 2.** *Suppose that $\mu(x) > 0$ for $x \in \mathbb{R}^d$. Then any stationary distribution $\rho(\theta)$ that satisfies $\rho(\theta) > 0$ for any $\theta \in \Theta$ has zero loss and satisfies the condition*

$$\int \sigma(x \cdot \theta)(\rho(\theta) - \rho_*(\theta))\mathrm{d}\theta = 0. \tag{4}$$

*Proof.* By introducing the operator $\Sigma$ with kernel given by $\sigma(x \cdot \theta)$, one can write the loss as $E(\rho) = \frac{1}{2}\|\Sigma(\rho - \rho_*)\|_\mu^2$ and the functional derivative as

$$\frac{\delta E}{\delta \rho} = \Sigma^\mathsf{T} \mu \Sigma(\rho - \rho_*),$$

where $\mu$ is considered to the operator of multiplying with $\mu(x)$ at each $x$. At a stationary distribution $\rho$, since $\rho_t \equiv 0$, $\mathrm{div}_\theta \left[ \rho \nabla_\theta \frac{\delta E}{\delta \rho} \right] = 0$ at any $\theta \in \Theta$. Taking inner product with $\frac{\delta E}{\delta \rho}$ gives

$$\left( \frac{\delta E}{\delta \rho} \right)^\mathsf{T} \mathrm{div}_\theta \left[ \rho \nabla_\theta \frac{\delta E}{\delta \rho} \right] = 0.$$

Using the fact that $\mathrm{div}_\theta$ is the adjoint of $\nabla_\theta$, this states $\sqrt{\rho(\theta)} \nabla_\theta \frac{\delta E}{\delta \rho} = 0$, which further implies $\nabla_\theta \frac{\delta E}{\delta \rho} = 0$ as $\rho(\theta) > 0$. This means that $\frac{\delta E}{\delta \rho} = \Sigma^\mathsf{T} \mu \Sigma(\rho - \rho_*)$ as a function of $\theta$ is a constant. Taking inner-product of $\frac{\delta E}{\delta \rho}$ with $(\rho - \rho_*)(\theta)$ and using the fact that $\int (\rho - \rho_*)(\theta)\mathrm{d}\theta = 0$ gives

$$(\rho - \rho_*)^\mathsf{T} \Sigma^\mathsf{T} \mu \Sigma(\rho - \rho_*) = 0.$$

This implies zero loss and $\sqrt{\mu(x)} \Sigma(\rho - \rho_*) = 0$, which further implies $\Sigma(\rho - \rho_*) = 0$ since $\mu(x) > 0$. Writing $\Sigma(\rho - \rho_*) = 0$ in the integral form gives (4). $\square$

Below we consider the implication of Theorem 2 for a few commonly-used activation functions.

**Example 1.** *Consider the ReLU activation $\sigma(z) = \max(z, 0)$. It is convenient to use the polar coordinates $x \to (\hat{x}, |x|)$ and $\theta \to (\hat{\theta}, |\theta|)$, where $|\cdot|$ and $\hat{\cdot}$ denote the radial coordinate and the angular variables, respectively. Then (4) reads*

$$\int_{\hat{\theta}} \int_{|\theta|} |x||\theta|^d \sigma(\hat{x} \cdot \hat{\theta})(\rho(\hat{\theta}, |\theta|) - \rho_*(\hat{\theta}, |\theta|))\mathrm{d}\hat{\theta}d|\theta| = 0,$$

*where the factor $|\theta|^d$ comes from the homogeneity of ReLU as well as the Jacobian from the polar coordinate transformation. If the operator with kernel $(\sigma(\hat{x} \cdot \hat{\theta}))_{\hat{x}, \hat{\theta} \in \mathbb{S}^{d-1}}$ has a trivial null space, then one can conclude that the stationary distribution $\rho(\hat{\theta}, |\theta|)$ satisfies for any $\hat{\theta} \in \mathbb{S}^{d-1}$*

$$\int_{|\theta|} |\theta|^d (\rho(\hat{\theta}, |\theta|) - \rho_*(\hat{\theta}, |\theta|))d|\theta| = 0.$$

**Example 2.** *Consider for simplicity the 1D case $d = 1$. When $\sigma(z)$ is an even function such as the Gaussian or quadratic activations, (4) states that $\rho(\theta)$ is a stationary distribution if and only if the difference $(\rho - \rho_*)(\theta)$ is an odd function.*

*When $\sigma(z)$ is an odd function such as tanh or arctan, (4) states that $\rho$ is a stationary distribution if and only if the difference $(\rho - \rho_*)(\theta)$ is an even function. In addition, when $\sigma(z)$ is the sum of an odd function and a constant (for example, like the Sigmoid function), (4) is equivalent to $(\rho - \rho_*)(\theta)$ being even, since the integral of $(\rho - \rho_*)(\theta)$ is zero.*

*Finally, the softplus function $\sigma(z) = \ln(1 + e^z)$, which is a smooth approximation of ReLU. (4) states that*

$$\int \ln(1 + e^{\theta x})(\rho - \rho_*)(\theta)\mathrm{d}\theta = 0.$$

*Let us assume for simplicity that both $\rho(\theta)$ and $\rho_*(\theta)$ decay faster than $\theta^{-2}$ at large $\theta$ and define $f(\theta)$ as the anti-derivative of $(\rho - \rho_*)(\theta)$. Integrating by parts for (4) results in*

$$\int \frac{1}{1 + e^{-\theta x}} f(\theta)\mathrm{d}\theta = 0.$$

*Noticing that the Sigmoid function $\frac{1}{1+e^{-\theta x}}$ is a sum of an odd function and a constant, we conclude that* (4) *is equivalent to* $f(\theta)$ *being even and mean-zero. In terms of the difference* $(\rho - \rho_*)(\theta)$*, these two conditions means that* $(\rho - \rho_*)(\theta)$ *is odd and has a vanishing first moment*

$$\int \theta \cdot (\rho - \rho_*)(\theta) \mathrm{d}\theta = 0.$$

## 2.2 CONVERGENCE SPEED ISSUE

One interesting empirical observation from training neural networks in the teacher-student setting is that the convergence to a teacher parameter with a large weight is much faster than the speed to a teacher parameter with a small weight. More specifically, consider a teacher network given by

$$\rho_*(\theta) = \alpha_1 \delta_{c_1}(\theta) + \alpha_2 \delta_{c_2}(\theta),$$

where $c_i$ are the parameter values in $\mathbb{R}^d$ and the weights $\alpha_i$ satisfy $\alpha_1 \gg \alpha_2 > 0$ and $\alpha_1 + \alpha_2 = 1$. This can either be realized as a general two-layer network of two hidden units with weights $\alpha_1$ and $\alpha_2$ or an equal-weight network (1) of $\alpha_1 N$ units with parameter $c_1$ and $\alpha_2 N$ units with parameter $c_2$. After an initial short period, the evolution of the student parameters decouple into two nearly independent problems: an $\alpha_1$ fraction of the $N$ student parameters converges to $c_1$, while an $\alpha_2$ fraction of the student parameters converges to $c_2$. The observed behavior is that the convergence to $c_1$ is much faster than the convergence to $c_2$.

The mean field PDE allows us to understand this phenomenon using scale analysis. After the initial stage, the PDE $\rho_t = \mathrm{div}_\theta \left[ \rho \nabla_\theta \left( \Sigma^\mathsf{T} \mu \Sigma (\rho - \rho_*) \right) \right]$ is effectively decoupled into two equations

$$\rho_t^1 = \mathrm{div}_\theta \left[ \rho^1 \nabla_\theta \left( \Sigma^\mathsf{T} \mu \Sigma (\rho^1 - \rho_*^1) \right) \right], \quad \text{where } \rho_*^1 = \alpha_1 \delta_{c_1}(\theta),$$

$$\rho_t^2 = \mathrm{div}_\theta \left[ \rho^2 \nabla_\theta \left( \Sigma^\mathsf{T} \mu \Sigma (\rho^2 - \rho_*^2) \right) \right], \quad \text{where } \rho_*^2 = \alpha_2 \delta_{c_2}(\theta).$$

The key observation is that the *quadratic* dependence on $\rho$ on the right hand side is the reason of different convergence speeds. To see this, let us divide these two equations by $\alpha_1$ and $\alpha_2$, respectively,

$$\left( \frac{\rho^1}{\alpha_1} \right)_t = \alpha_1 \, \mathrm{div}_\theta \left[ \frac{\rho^1}{\alpha_1} \nabla_\theta \left( \Sigma^\mathsf{T} \mu \Sigma \left( \frac{\rho^1}{\alpha_1} - \frac{\rho_*^1}{\alpha_1} \right) \right) \right], \quad \text{where } \frac{\rho_*^1}{\alpha_1} = \delta_{c_1}(\theta),$$

$$\left( \frac{\rho^2}{\alpha_2} \right)_t = \alpha_2 \, \mathrm{div}_\theta \left[ \frac{\rho^2}{\alpha_2} \nabla_\theta \left( \Sigma^\mathsf{T} \mu \Sigma \left( \frac{\rho^2}{\alpha_2} - \frac{\rho_*^2}{\alpha_2} \right) \right) \right], \quad \text{where } \frac{\rho_*^2}{\alpha_2} = \delta_{c_2}(\theta).$$

These two are the same equations for normalized densities $\frac{\rho^1}{\alpha_1}$ and $\frac{\rho^2}{\alpha_2}$, except the coefficients $\alpha_1$ and $\alpha_2$ at the beginning of the right hand sides. The difference in these two coefficients clearly demonstrates that the first equation converges $\alpha_1/\alpha_2$ times faster than the second one.

## 3 RESNET MODELS IN TEACHER-STUDENT SETTING

Consider a simple ResNet model with input $x \in \mathbb{R}^d$ (see Figure 1 left)

$$X^0 = x, \quad X^{\ell+1} = X^\ell + \frac{1}{L} \sigma(\theta^\ell X^\ell), \quad 0 \le \ell < L,$$

where $X^\ell \in \mathbb{R}^d$, the matrix parameter $\theta^\ell \in \Theta \subset \mathbb{R}^{d \times d}$, and the activation $\sigma$ is applied entry-wise to each component. By introducing $\boldsymbol{\theta} = (\theta^0, \dots, \theta^{L-1})$, we denote the final result at level $L$ by $X_{\boldsymbol{\theta}}^L$ in order to emphasize the dependence on $\boldsymbol{\theta}$. Given a function $y : \mathbb{R}^d \to \mathbb{R}^d$, the training problem searches for the parameters $\boldsymbol{\theta} = (\theta^0, \dots, \theta^{L-1}) \in \Theta^L$ in order to minimize, say, the quadratic loss

$$E(\boldsymbol{\theta}) = \frac{1}{2} \int_x \left| X_{\boldsymbol{\theta}}^L(x) - y(x) \right|^2 \mu(x) \mathrm{d}x = \frac{1}{2} \left\| X_{\boldsymbol{\theta}}^L(x) - y(x) \right\|_\mu^2.$$

When optimized with GD, the parameter vector $\boldsymbol{\theta}$ is updated at each step with

$$\boldsymbol{\theta}[k+1] = \boldsymbol{\theta}[k] - \alpha \frac{\partial E}{\partial \boldsymbol{\theta}}(\boldsymbol{\theta}[k])$$

(or with an extra noise term for SGD). Instead of directly working with this vanilla ResNet, we introduce two extended ResNet models in the rest of this section and derive a mean field equation for each of them.

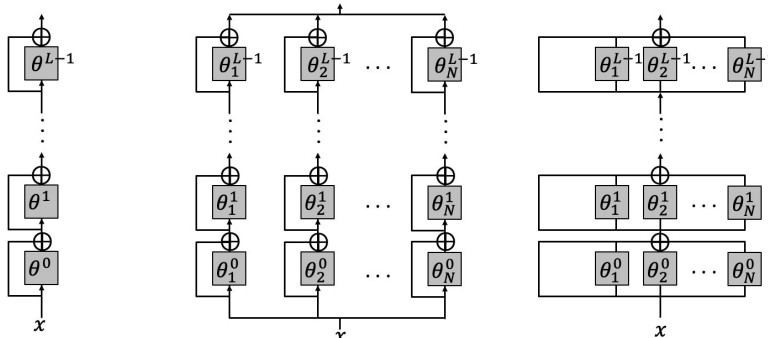

Figure 1: ResNet models. Left: vanilla ResNet. Middle: ensemble-ResNet. Right: wide-ResNet. Mean field models for the ensemble-ResNet and wide-ResNet models are discussed in Section 3.

### 3.1 ENSEMBLE-RESNET

**Setup.** The first model, called *ensemble-ResNet*, averages over a group of $N$ ResNets with independent parameters (see Figure 1 middle). Its $j$-th ResNet has the same structure as the vanilla ResNet described above and contains parameters $\boldsymbol{\theta}_j = (\theta_j^0, \ldots, \theta_j^{L-1}) \in \Theta^L$. The output of the ensemble-ResNet is $\frac{1}{N} \sum_{j=1}^N X_{\boldsymbol{\theta}_j}^L(x)$ and the loss function is

$$E(\boldsymbol{\theta}_1, \ldots, \boldsymbol{\theta}_N) = \frac{1}{2} \left\| \frac{1}{N} \sum_{j=1}^N X_{\boldsymbol{\theta}_j}^L(x) - y(x) \right\|_\mu^2 .$$

The GD/SGD algorithm with learning rate $\alpha$ takes the following step for each $i$

$$\boldsymbol{\theta}_i[k+1] = \boldsymbol{\theta}_i[k] - \alpha \int_x \frac{1}{N} \nabla_{\boldsymbol{\theta}_i} X_{\boldsymbol{\theta}_i[k]}^L(x)^\mathsf{T} \left( \frac{1}{N} \sum_{j=1}^N X_{\boldsymbol{\theta}_j[k]}^L(x) - y(x) \right) \mu(x) \mathrm{d}x.$$

When $\alpha/N$ is sufficiently small, the dynamics of $\boldsymbol{\theta}_i[k]$ can be approximated by an ODE system $\boldsymbol{\theta}_i(t)$ by identifying $t = k\alpha/N$

$$\frac{\mathrm{d}\boldsymbol{\theta}_i}{\mathrm{d}t} = - \int_x \nabla_{\boldsymbol{\theta}_i} X_{\boldsymbol{\theta}_i}^L(x)^\mathsf{T} \left( \frac{1}{N} \sum_{j=1}^N X_{\boldsymbol{\theta}_j}^L(x) - y(x) \right) \mu(x) \mathrm{d}x.$$

Viewing $\boldsymbol{\theta}_i(t) \equiv (\theta_i^0(t), \ldots, \theta_i^{L-1}(t))$ as particles in $\Theta^L$ and introducing the distribution $\rho(t, \boldsymbol{\theta}) = \frac{1}{N} \sum_{j=1}^N \delta_{\boldsymbol{\theta}_j(t)}(\boldsymbol{\theta})$, we rewrite $E(\boldsymbol{\theta}_1, \ldots, \boldsymbol{\theta}_N)$ as $E(\rho) \equiv \frac{1}{2} \left\| \int_{\boldsymbol{\theta}} X_{\boldsymbol{\theta}}^L(x) \rho(\boldsymbol{\theta}) \mathrm{d}\boldsymbol{\theta} - y(x) \right\|_\mu^2$. The mean field equation of $\rho(t, \boldsymbol{\theta})$ takes the same form as (3)

$$\rho_t(\boldsymbol{\theta}) = \mathrm{div}_{\boldsymbol{\theta}} \left[ \rho \nabla_{\boldsymbol{\theta}} \frac{\delta E}{\delta \rho} \right],$$

which is a gradient flow of quadratic loss $E(\rho)$ under the Wasserstein metric on the densities $\rho(\boldsymbol{\theta})$ on the product space $\Theta^L$.

**Stationary distribution in teacher-student setting.** In the teacher-student setting, the function $y(x)$ from $\mathbb{R}^d$ to $\mathbb{R}^d$ is assumed to take the form $y(x) = \int_{\boldsymbol{\theta}} X_{\boldsymbol{\theta}}^L(x) \rho_*(\boldsymbol{\theta}) \mathrm{d}\boldsymbol{\theta}$, for a certain distribution $\rho_*$ on the space $\Theta^L$. This include the special case that $y(x)$ can be represented by a vanilla ResNet described above. More precisely, assuming that

$$X^0 = x, \quad X^{\ell+1} = X^\ell + \frac{1}{L} \sigma(\theta_*^\ell X^\ell), \quad y(x) = X^L(x),$$

then $y(x)$ corresponds to $\rho_*(\boldsymbol{\theta}) = \delta_{\boldsymbol{\theta}_*}(\boldsymbol{\theta})$ with $\boldsymbol{\theta}_* \equiv (\theta_*^0, \ldots, \theta_*^{L-1}) \in \Theta^L$.

**Theorem 3.** *Suppose that $\mu(x) > 0$ for $x \in \mathbb{R}^d$. Then any stationary distribution $\rho(\boldsymbol{\theta})$ that satisfies $\rho(t, \boldsymbol{\theta}) > 0$ for any $t > 0$ and $\boldsymbol{\theta} \in \Theta^L$ has zero loss and satisfies the condition, for each $x$,*

$$\int_{\boldsymbol{\theta}} X_{\boldsymbol{\theta}}^L(x)(\rho(\boldsymbol{\theta}) - \rho_*(\boldsymbol{\theta}))\mathrm{d}\boldsymbol{\theta} = 0. \tag{5}$$

The proof of the theorem follows the same argument as Theorem 2. One potential criticism of the ensemble-ResNet model is that the space $\Theta^L \subset (\mathbb{R}^{d \times d})^L$ can be very high-dimensional even for moderate values of $L$. Therefore, the number of branches $N$ required in order for the mean field model to be accurate can be quite large.

### 3.2 WIDE-RESNET

In order to overcome the exponential grow of the space $\Theta^L$ in the ensemble-ResNet model, we consider a different ResNet model in this subsection. Instead of allowing for multiple independent branches, the *wide-ResNet* model sticks to a single path but increases the width of each residual block (see Figure 1 right). For an input $x \in \mathbb{R}^d$, the wide-ResNet evaluates

$$X^0 = x, \quad X^{\ell+1} = X^\ell + \frac{1}{LN} \sum_{j=1}^N \sigma(\theta_j^\ell X^\ell), \quad 0 \le \ell < L, \tag{6}$$

where the activation function $\sigma(z)$ is assumed to be smooth and vanishes at $z = 0$. By introducing the parameter group $\boldsymbol{\theta}^\ell = (\theta_1^\ell, \dots, \theta_N^\ell)$ for each level $\ell$, we write the final result of (6) as $X_{\boldsymbol{\theta}^0, \dots, \boldsymbol{\theta}^{L-1}}^L(x)$ and the quadratic loss function as

$$E(\boldsymbol{\theta}^0, \dots, \boldsymbol{\theta}^{L-1}) = E(\theta_1^0, \dots, \theta_N^0, \dots, \theta_1^{L-1}, \dots, \theta_N^{L-1}) = \frac{1}{2} \left\| X_{\boldsymbol{\theta}^0, \dots, \boldsymbol{\theta}^{L-1}}^L(x) - y(x) \right\|_\mu^2.$$

The GD/SGD algorithm with learning rate $\alpha$ takes the form

$$\theta_i^\ell[k+1] = \theta_i^\ell[k] - \alpha \int_x \nabla_{\theta_i^\ell} X_{\boldsymbol{\theta}^0[k], \dots, \boldsymbol{\theta}^{L-1}[k]}^L(x)^\mathsf{T} \left( X_{\boldsymbol{\theta}^0[k], \dots, \boldsymbol{\theta}^{L-1}[k]}^L(x) - y(x) \right) \mu(x)\mathrm{d}x,$$

where $\nabla_{\theta_i^\ell} X_{\boldsymbol{\theta}^0[k], \dots, \boldsymbol{\theta}^{L-1}[k]}^L(x)$ contains a factor $\frac{1}{LN}$. When the stepsize $\alpha$ is sufficiently small, the dynamics of $\theta_i^\ell[k]$ can approximated with an ODE system

$$\frac{\mathrm{d}\theta_i^\ell}{\mathrm{d}t} = -\int_x \nabla_{\theta_i^\ell} X_{\boldsymbol{\theta}^0, \dots, \boldsymbol{\theta}^{L-1}}^L(x)^\mathsf{T} \left( X_{\boldsymbol{\theta}^0, \dots, \boldsymbol{\theta}^{L-1}}^L(x) - y(x) \right) \mu(x)\mathrm{d}x.$$

By viewing $\theta_1^\ell(t), \dots, \theta_N^\ell(t)$ as interacting particles at each level $\ell$ and introducing $\rho^\ell(t, \theta^\ell) = \frac{1}{N} \sum_j \delta_{\theta_j^\ell(t)}(\theta^\ell)$ for each $\ell$, one can rewrite the wide-ResNet computation as

$$X^0 = x, \quad X^{\ell+1} = X^\ell + \frac{1}{L} \int \sigma(\theta^\ell X^\ell)\rho^\ell(\theta^\ell)\mathrm{d}\theta^\ell, \quad 0 \le \ell < L. \tag{7}$$

By taking the limit $N$ to infinity, the loss becomes $E(\rho^0, \dots, \rho^{L-1}) = \frac{1}{2} \left\| X_{\rho^0, \dots, \rho^{L-1}}^L(x) - y(x) \right\|_\mu^2$ (with the subscripts denoting the dependence on $\rho^0, \dots, \rho^{L-1}$) and the mean field equation for $\rho^\ell(t, \theta^\ell)$ for each level $\ell$ reads

$$\rho_t^\ell(\theta^\ell) = \mathrm{div}_{\theta^\ell} \left[ \rho^\ell(\theta^\ell) \nabla_{\theta^\ell} \frac{\delta E(\rho^0, \dots, \rho^{L-1})}{\delta \rho^\ell} \right]. \tag{8}$$

As $\sigma(z)$ is assumed to be smooth, expanding (7) recursively and applying the Taylor expansion (see Appendix 5.2 for a derivation) gives the following approximation

$$X^L \approx X^0 + \frac{1}{L} \sum_{a=0}^{L-1} \int \sigma(\theta X^0)\rho^a(\theta)\mathrm{d}\theta + \frac{1}{L^2} \sum_{b>a} \iint \nabla\sigma(\theta^b X^0)\theta^b \sigma(\theta^a X^0)\rho^b(\theta^b)\rho^a(\theta^a)\mathrm{d}\theta^a\mathrm{d}\theta^b + \dots \tag{9}$$

Furthermore, since $\sigma(z)$ vanishes at zero by assumption, this expansion has $L$ terms of order $O(\theta)$ each with weight $1/L$, $\binom{L}{2}$ terms of order $O(\theta^2)$ each with weight $1/L^2$, etc. When the domain $\Theta$ of the matrix parameter $\theta$ is bounded by a sufficiently small radius, the contribution from the $k$-linear terms is of order $O(1/k!)$ and hence decays rapidly with increasing values of $k$. Therefore, in the same parameter regime, it is sufficient to consider approximations of the first few orders.

**Linear approximation.** The linear approximation for $X^L$ in (9) leads to the following first-order approximation to the loss

$$E(\rho^0, \dots, \rho^{L-1}) \approx E_{(1)}(\rho^0, \dots, \rho^{L-1}) \equiv \frac{1}{2} \left\| x + \frac{1}{L} \sum_a \int \sigma(\theta x) \rho^a(\theta) \mathrm{d}\theta - y(x) \right\|_\mu^2.$$

Now take the limit of $L$ going to infinity and define $\rho(\theta, \varepsilon) = \sum_a \rho^a(\theta) \cdot \delta_{a/L}(\varepsilon)$ where $\varepsilon \in [0, 1]$ is a rescaled continuous variable associated with the depth of the ResNet. In this formal limit, the loss function becomes $E_{(1)}(\rho) \equiv \frac{1}{2} \left\| x + \iint \sigma(\theta x) \rho(\theta, \varepsilon) \mathrm{d}\theta \mathrm{d}\varepsilon - y(x) \right\|_\mu^2$. The mean field equation for $\rho(t, \theta, \varepsilon)$ takes the form

$$\rho_t(\theta, \varepsilon) = \mathrm{div}_\theta \left[ \rho(\theta, \varepsilon) \nabla_\theta \frac{\delta E_{(1)}}{\delta \rho} \right]. \tag{10}$$

Notice that this is a degenerate gradient flow with mobility only in the $\theta$ variable. In the teacher-student setting, i.e. $y(x) = x + \iint \sigma(\theta x) \rho_*(\theta, \varepsilon) \mathrm{d}\theta \mathrm{d}\varepsilon$ for some teacher distribution $\rho_*(\theta, \varepsilon)$, we can write compactly $E_{(1)}(\rho) = \frac{1}{2} \|\Sigma_1(\rho - \rho_*)\|_\mu^2$, where $\Sigma_1$ is the operator with kernel $\sigma(\theta x)$. Notice that $\varepsilon$ does not appear in the kernel explicitly.

**Theorem 4.** *Suppose that $\mu(x) > 0$ for $x \in \mathbb{R}^d$. Then any stationary distribution $\rho(\theta, \varepsilon)$ that satisfies $\rho(\theta, \varepsilon) > 0$ for any $\theta \in \Theta$ and $\varepsilon \in [0, 1]$ has zero loss and satisfies, at each $x \in \mathbb{R}^d$,*

$$\iint_\theta \sigma(\theta x) \left( \int \rho \mathrm{d}\varepsilon - \int \rho_* \mathrm{d}\varepsilon \right) \mathrm{d}\theta = 0. \tag{11}$$

The proof follows the argument of Theorem 2. It is not surprising that the constraint (11) is formulated in terms of the marginal of $\rho$, since the parameters at different levels become almost commutative in the small $\theta$ regime.

**Quadratic approximation.** The quadratic approximation for $X^L$ in (9) leads to the following second-order approximation to the loss

$$E_{(2)}(\rho^0, \dots, \rho^{L-1}) \equiv \frac{1}{2} \Bigg\| x + \frac{1}{L} \sum_{a=0}^{L-1} \int \sigma(\theta x) \rho^a(\theta) \mathrm{d}\theta - y(x)$$

$$+ \frac{1}{L^2} \sum_{b>a} \iint \nabla \sigma(\theta^b X^0) \theta^b \sigma(\theta^a X^0) \rho^b(\theta^b) \rho^a(\theta^a) \mathrm{d}\theta^a \mathrm{d}\theta^b - y(x) \Bigg\|_\mu^2.$$

Taking the large $L$ limit and introducing $\rho(\theta, \varepsilon) \equiv \sum_a \rho^a(\theta) \cdot \delta_{a/L}(\varepsilon)$ gives rise to the loss function

$$E_{(2)}(\rho) \equiv \frac{1}{2} \left\| x + \int \sigma(\theta x) \rho(\theta, \varepsilon) \mathrm{d}\theta \mathrm{d}\varepsilon + \iint_{\varepsilon'>\varepsilon} \nabla \sigma(\theta' x) \theta' \sigma(\theta x) \rho(\theta', \varepsilon') \rho(\theta, \varepsilon) \mathrm{d}\theta' \mathrm{d}\theta \mathrm{d}\varepsilon' \mathrm{d}\varepsilon - y(x) \right\|_\mu^2$$

in terms of $\rho(\theta, \varepsilon)$ and the mean field equation takes the same form as (10) except with $E_{(1)}$ replaced with $E_{(2)}$. In the teacher-student setting, there exists some $\rho_*(\theta, \varepsilon)$ such that

$$y(x) = x + \int \sigma(\theta x) \rho_*(\theta, \varepsilon) \mathrm{d}\theta \mathrm{d}\varepsilon + \iint_{\varepsilon'>\varepsilon} \nabla \sigma(\theta' x) \theta' \sigma(\theta x) \rho_*(\theta', \varepsilon') \rho_*(\theta, \varepsilon) \mathrm{d}\theta' \mathrm{d}\theta \mathrm{d}\varepsilon' \mathrm{d}\varepsilon.$$

By introducing a new operator $\Sigma_2(\rho \otimes \rho) = \iint_{\varepsilon'>\varepsilon} \nabla \sigma(\theta' x) \theta' \sigma(\theta x) \rho_*(\theta', \varepsilon') \rho_*(\theta, \varepsilon) \mathrm{d}\theta' \mathrm{d}\theta \mathrm{d}\varepsilon' \mathrm{d}\varepsilon$, we can write it more compactly as

$$E_{(2)}(\rho) = \frac{1}{2} \left\| \Sigma_1(\rho - \rho_*) + \Sigma_2(\rho \otimes \rho - \rho_* \otimes \rho_*) \right\|_\mu^2.$$

Unfortunately, since $E_{(2)}(\rho)$ is quartic in $\rho$, a stationary distribution might be a local minimum even with the extra assumption $\rho(\theta, \varepsilon) > 0$. On the other hand, the explicit and simple form of $E_{(2)}(\rho)$ might allow multiple runs with different initial condition $\rho(0, \theta)$ to identify the $\rho_*$ more efficiently.

## 4 DISCUSSIONS

This paper discusses necessary conditions for the stationary distributions of mean field models for the two-layer neural networks as well as two extended ResNet models. A lot of interesting questions remain to be addressed concerning the ResNet. The first, and probably most important, question is that whether it is possible to obtain a mean field model for the vanilla ResNet by treating the depth $L$ as the large parameter going to infinity. This in fact has been the original motivation of this paper and it still remains unresolved. Second, in the wide-ResNet model, the resulting PDE is degenerate as there is no diffusion in the $\varepsilon$ variable. Would it be possible to include the diffusion in $\varepsilon$ by allowing parameters to jump between adjacent layers? If this is possible, the next step is to formulate a mean field model with densities over some "architecture" space for neural architecture search (Zoph & Le, 2016; Zoph et al., 2018; Elsken et al., 2018; Wistuba et al., 2019).

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

## 5  APPENDIX

### 5.1  TWO-LAYER NETWORK WITH VARYING WEIGHTS

In this subsection, we discuss the mean field equation for the general two-layer network model with trainable weights

$$\frac{1}{N}\sum_{j=1}^{N}w_j\sigma(x\cdot\theta_j),$$

where $w_j\in W\subset\mathbb{R}$ for each $j$. The quadratic loss $E(\theta_1,w_1,\ldots,\theta_N,w_N)$ is given by

$$E(\theta_1,w_1,\ldots,\theta_N,w_N)=\frac{1}{2}\left\|\frac{1}{N}\sum_{j=1}^{N}w_j\sigma(x\cdot\theta_j)-y(x)\right\|_{\mu}.$$

GD or SGD with sufficiently small step-size can be modeled by the ODE system

$$\frac{d\theta_i(t)}{dt}=-\int_x\left(\frac{1}{N}\sum_{j=1}^{N}w_j\sigma(x\cdot\theta_j(t))-y(x)\right)w_i(t)\sigma'(x\cdot\theta_i(t))x\mu(x)\mathrm{d}x,$$

$$\frac{dw_i(t)}{dt}=-\int_x\left(\frac{1}{N}\sum_{j=1}^{N}w_j\sigma(x\cdot\theta_j(t))-y(x)\right)\sigma(x\cdot\theta_i(t))\mu(x)\mathrm{d}x.$$

Viewing $(\theta_i(t),w_i(t))$ as interacting particles in $\Theta\times W$ and considering the distribution

$$\rho(t,\theta,w)\equiv\frac{1}{N}\sum_{j=1}^{N}\delta_{\theta_j(t),w_j(t)}(\theta,w),$$

one can write $\frac{1}{N}\sum_j w_j\sigma(x\cdot\theta_j)-y(x)$ as $\int w\sigma(x\cdot\theta)\rho(\theta,w)\mathrm{d}\theta\mathrm{d}w$ and $E(\theta_1,w_1,\ldots,\theta_N,w_N)$ as

$$E(\rho)\equiv\frac{1}{2}\left\|\int w\sigma(x\cdot\theta)\rho(\theta,w)\mathrm{d}\theta\mathrm{d}w-y(x)\right\|_{\mu}^{2}.$$

The mean field equation for $\rho(t,\theta,w)$ takes the form

$$\rho_t=\mathrm{div}_{\theta w}\left[\rho\nabla_{\theta w}\frac{\delta E}{\delta\rho}\right],\tag{12}$$

which is the gradient flow of $E(\rho)$ under the Wasserstein metric on the densities $\rho(\theta,w)$. In the teacher-student setting, i.e., $y(x)=\int_{\theta w}w\sigma(x\cdot\theta)\rho_*(\theta,w)\mathrm{d}\theta\mathrm{d}w$ for some teacher parameter distribution $\rho_*(\theta,w)$,

$$E[\rho]=\frac{1}{2}\int_x\left(\int_{\theta w}w\sigma(x\cdot\theta)(\rho(\theta,w)-\rho_*(\theta,w)\mathrm{d}\theta\right)^2\mathrm{d}x.$$

The following theorem characterizes the stationary distribution of (12).

**Theorem 5.** *Suppose that $\mu(x)>0$ for $x\in\mathbb{R}^d$. Then any stationary distribution $\rho(\theta,w)$ that satisfies $\rho(\theta,w)>0$ for any $\theta\in\Theta$ and $w\in W$ has zero loss and satisfies the condition*

$$\int w\sigma(x\cdot\theta)(\rho(\theta,w)-\rho_*(\theta,w))\mathrm{d}\theta\mathrm{d}w=0.\tag{13}$$

*Proof.* By introducing the operator $\Sigma$ with kernel given by $w\sigma(x\cdot\theta)$, one can write the loss $E(\rho)=\frac{1}{2}\|\Sigma(\rho-\rho_*)\|_{\mu}^2$ and

$$\frac{\delta E}{\delta\rho}=\Sigma^{\mathsf{T}}\mu\Sigma(\rho-\rho_*),$$

where $\mu$ is considered to the operator of multiplying with $\mu(x)$ at each $x$. At a stationary distribution $\rho$, since $\rho_t \equiv 0$, $\mathrm{div}_{\theta w}\left[\rho\nabla_{\theta w}\frac{\delta E}{\delta\rho}\right] = 0$ at any $\theta \in \Theta$ and $w \in W$. Taking inner product with $\frac{\delta E}{\delta\rho}$ gives

$$\left(\frac{\delta E}{\delta\rho}\right)^{\mathsf{T}}\mathrm{div}_{\theta w}\left[\rho(\theta, w)\nabla_{\theta w}\frac{\delta E}{\delta\rho}\right] = 0.$$

Using the fact that $\mathrm{div}_\theta$ is the adjoint of $\nabla_{\theta w}$, this means $\sqrt{\rho(\theta, w)}\nabla_{\theta w}\frac{\delta E}{\delta\rho} = 0$, which further implies $\nabla_{\theta w}\frac{\delta E}{\delta\rho} = 0$ as $\rho(\theta, w) > 0$. This means that $\frac{\delta E}{\delta\rho} = \Sigma^{\mathsf{T}}\mu\Sigma(\rho - \rho_*)$ as a function of $\theta$ and $w$ is constant. Taking inner-product of $\frac{\delta E}{\delta\rho}$ with $(\rho - \rho_*)(\theta, w)$, one arrives at

$$(\rho - \rho_*)^{\mathsf{T}}\Sigma^{\mathsf{T}}\mu\Sigma(\rho - \rho_*) = 0,$$

using the fact that $\int(\rho - \rho_*)(\theta, w)\mathrm{d}\theta\mathrm{d}w = 0$. This implies zero loss and

$$\sqrt{\mu(x)}\Sigma(\rho - \rho_*) = 0,$$

which further implies $\Sigma(\rho - \rho_*) = 0$ since $\mu(x) > 0$. Writing the last statement explicitly in the integral form gives (13). $\qquad\square$

**Example 3.** *Let us consider the ReLU activation $\sigma(z) = \max(z, 0)$. Using the polar coordinates $x \to (\hat{x}, |x|)$ and $\theta \to (\hat{\theta}, |\theta|)$, (13) reads*

$$\int_{\hat{\theta}}\int_{|\theta|}\int_w |x||\theta|^d w\sigma(\hat{x}\cdot\hat{\theta})(\rho(\hat{\theta}, |\theta|, w) - \rho_*(\hat{\theta}, |\theta|, w))\mathrm{d}\hat{\theta}\mathrm{d}|\theta|\mathrm{d}w = 0.$$

*If the operator with the kernel $(\sigma(\hat{x}\cdot\hat{\theta}))_{\hat{x},\hat{\theta}\in\mathbb{S}^{d-1}}$ has a trivial null space, then one can conclude that that for each $\hat{\theta}$*

$$\int_{|\theta|}\int_w |\theta|^d w\left(\rho(\hat{\theta}, |\theta|, w) - \rho_*(\hat{\theta}, |\theta|, w)\right)\mathrm{d}|\theta|\mathrm{d}w = 0$$

*that at the stationary distribution $\rho$.*

## 5.2   DERIVATION OF (9)

The computation follows the definition in (7) and uses Taylor expansion. At the first level,

$$X^1 = X^0 + \frac{1}{L}\int_{\theta^0}\sigma(\theta^0 X^0)\rho^0(\theta^0)\mathrm{d}\theta^0.$$

At the second level,

$$X^2 = X^1 + \frac{1}{L}\int_{\theta^1}\sigma(\theta^1 X^1)\rho^1(\theta^1)\mathrm{d}\theta^1$$
$$= X^0 + \frac{1}{L}\int_{\theta^0}\sigma(\theta^0 X^0)\rho^0(\theta^0)\mathrm{d}\theta^0 + \frac{1}{L}\int_{\theta^1}\sigma\left(\theta^1\left(X^0 + \frac{1}{L}\int_{\theta^0}\sigma(\theta^0 X^0)\rho^0(\theta^0)\mathrm{d}\theta^0\right)\right)\rho^1(\theta^1)\mathrm{d}\theta^1.$$

Recalling that $\sigma$ is assumed to be smooth near the origin, a Taylor expansion for $\sigma$ in the last term

$$X^2 \approx X^0 + \frac{1}{L}\int_{\theta^0}\sigma(\theta^0 X^0)\rho^0(\theta^0)\mathrm{d}\theta^0 + \frac{1}{L}\int_{\theta^1}\sigma(\theta^1 X^0)\rho^1(\theta^1)\mathrm{d}\theta^1$$
$$+ \frac{1}{L^2}\int_{\theta^1}\nabla\sigma(\theta^1 X^0)\theta^1\left(\int_{\theta^0}\sigma(\theta^0 X^0)\rho^0(\theta^0)\mathrm{d}\theta^0\right)\rho^1(\theta^1)\mathrm{d}\theta^1 + \text{h.o.t.}$$

Iterating this until the $L$-th level gives rise to

$$X^L \approx X^0 + \frac{1}{L}\sum_{a=0}^{L-1}\int\sigma(\theta X^0)\rho^a(\theta)\mathrm{d}\theta + \frac{1}{L^2}\sum_{b>a}\iint\nabla\sigma(\theta^b X^0)\theta^b\sigma(\theta^a X^0)\rho^b(\theta^b)\rho^a(\theta^a)\mathrm{d}\theta^a\mathrm{d}\theta^b + \text{h.o.t.}$$

