# OpenReview forum: "Mean Field Models for Neural Networks in Teacher-student Setting"
_ICLR.cc/2020/Conference — Reject_

### Official Review · AnonReviewer3 · 2019-10-22
**Official Blind Review #3**

**Rating:** 3

**Review:**

The paper studies the dynamics of neural network in the Teacher-Student setting, using the approach pionnered in the last few years.

Concerning the presentation and the review of other works, I am a bit surprise that the "teacher-student" setting appears out of the blue, without any references to any paper. This is not a new area! Especially when it concerns learning with SGD, these were the subject of intense studies, especially in the 80s, see for instance:
* Saad & Sola On-line learning in soft committee machines, '95
* Exact solution for on-line learning in multilayer neural networks '95
or the book "On-Line Learning in Neural Networks", By David Saad, with contribution of Botou, etc...
* Many of these results were proven rigorously recently in "Dynamics of stochastic gradient descent for two-layer neural networks in the teacher-student setup" by Goldt et al,

There is a difference with the current formulation: in those papers, both the teacher AND student had a finite-size second (or more) layer, while here, one is working in the mean-field regime where the student is infinity wide. This is indeed a different situation, where the system can be much more "over-parametrized". But this does not mean that the subject is "terra incognita".

There are three main sections in the paper, discussing the results.

* The first result is a theorem that, if I understand correctly, says that the stationary distribution of gradient descent on the population loss (i.e. the test error) a necessary condition for the stationary distribution is that it has zero generalisation error (Eq. 4). That seems like an incremental-step compared to previous results that write down the PDE (the four mean-field papers from last year) and it looks very similar to results that show gradient descent provably converges to global optima etc. I am not sure I see the importance of the result. Also, it is not clear "how long" SGD should run to reach such a point and this regime might be entirely out of interest.

* Sec. 2.2 also discusses the difference in learning to teacher nodes with large or small weights, resp. Again, this is well-known in the asymptotic limit and rather unsurprising. This is also discussed in, e.g.  in Saxe et al "Exact solutions to the nonlinear dynamics of learning in deep linear neural networks" at least in the linear case.

* The extension to ResNets is definitely more interesting. The authors write down mean-field equations for two models, and prove, if I understand correctly, that it is a necessary condition to recover the teacher weights to generalise perfectly, which, as I said above, seems unsurprising.

In the end, given the paper is not discussing the relevant literature in teacher-student setting, and that I (perphaps wrongly)
 do not find the results surprising enough, I would not support acceptance in ICLR.



**Experience Assessment:**

I have read many papers in this area.

**Review Assessment: Checking Correctness Of Derivations And Theory:**

I did not assess the derivations or theory.

**Review Assessment: Checking Correctness Of Experiments:**

I did not assess the experiments.

**Review Assessment: Thoroughness In Paper Reading:**

I read the paper at least twice and used my best judgement in assessing the paper.

---

> ### Author Response · Authors · 2019-11-07
> **Reply to Review 3**
>
> The authors were unfamiliar with the literature on teacher-student networks. As the reviewer pointed out, the difference is that in our setting the student network has infinite width.
>
> The reviewer raised the question ``"how long" SGD should run to reach such a point''? In fact, this is a hard question for the type of nonlinear PDEs in mathematical analysis. As far as we know, there is no definite result on the convergence rate. The dynamics can be stuck at stationary points that are not the global minimum. There has been some work to improve convergence by adding extra noise or including a birth-death process. However, these approaches address a different PDE than the original mean-field equation.
>
> We were not aware of the paper Saxe et al ``Exact solutions to the nonlinear dynamics of learning in deep linear neural networks''. We plan to cite it in the revision. However, we would like to mention that this paper concerns deep linear network while our network architecture has nonlinear activation functions.

---

### Official Review · AnonReviewer2 · 2019-10-22
**Official Blind Review #2**

**Rating:** 3

**Review:**

This paper studies the mean fields limit of neural networks and extends the mean field limit treatment to resnets. The paper seems to be presenting non-trivial formal results but I could not really understand what are their specific consequences of interest. What are the concrete insights that the results of this paper bring? Even as a theoretician, I did not really understand what should I take from this paper. As such I cannot recommend its acceptance. But it is well plausible that with better explanations I will understand its value and change my mind. Some more concrete questions/issues follow:

The introduction several times states: "First, for the two-layer networks in the teacher-student setting, we characterize for the first time the
necessary condition for a stationary distribution of the mean field PDE and show how it relates to the parameters of the teacher network. We also discuss how different activation functions influence the results." The outline states: "Section 2 concerns the mean field description of two-layer networks and provides necessary conditions for stationary distributions."

But where is this discussion of necessary condition? I cannot find a single mention of "necessary" in section 2 that only includes formal derivations and statements. Can the result of Section 2 be translated to somewhat more general audience? What is the take-home message there?

Where is the discussion on the role of the activation function? I see the two examples stating the formal result for two different activations, but again I am unable to understand what should one take from this?

I could understand the result of section 2.2. about stronger weight-teachers being learned first. But this is completely intuitive in the same way as independent components corresponding to larger eigenvalues would get learned first in PCA. See also
similar conclusions in works on so-called "INCREMENTAL LEARNING" in Andrew M Saxe, James L McClelland, and Surya Ganguli. Exact solutions to the nonlinear dynamics of learning in deep linear neural networks. arXiv preprint arXiv:1312.6120, 2013. And subsequent results. So this alone does not seem very novel result.

As I had hard time understanding the section two on the simpler feedforward case, Section 3 was even less clear to me. How does what is done here compare to what is known on resnets and what should somebody interested in resnets (but not specifically in the mean field equations) take out from this. Some non-technical summary of the findings is seriously missing in this paper.

Comments that are less fundamental to the overall understanding:

** The mean field treatment was also extended to the multi-layer case in: https://arxiv.org/abs/1906.00193

** The paper present nice account on previous results involving the mean field limit. The manuscript should also discuss the long line of papers analyzing the teacher student setting on one-hidden-layer neural networks. After-all this seems to be the main object of study here so the results only make sence presented against what was previously known. Notably the case of eq. (1) where the 2nd layer weights are fixed to one is a model called the committee machine widely considered in previous literature (in the case of finitely wide one-hidden-layer network).

The (non-exhaustive) list of paper on the teacher-student setting is (more references are in those papers):

-- The teacher-student setting was introduced in Garder, Derrida'83 (model B, https://iopscience.iop.org/article/10.1088/0305-4470/22/12/004/pdf).

-- In the classical textbook on neural networks Engel, Andreas, and Christian Van den Broeck. Statistical mechanics of learning. Cambridge University Press, 2001, there is a rather detailed account of many results on the setting from 80s and 90s.

-- Notably the plain SGD was analyzed via ODEs for the teacher-student setting in classical papers: David Saad and Sara A Solla. On-line learning in soft committee machines. Physical Review E, 52 (4):4225, 1995.
David Saad and Sara A Solla. Dynamics of on-line gradient descent learning for multilayer neural networks. In Advances in neural information processing systems, pp. 302–308, 1996.

-- This line of work was recently extended with a focus on the overparametrized regime (but not infinitely wide) in: Dynamics of stochastic gradient descent for two-layer neural networks in the teacher-student setup, Sebastian Goldt, Madhu S. Advani, Andrew M. Saxe, Florent Krzakala, Lenka Zdeborová, NeurIPS'19.

-- There even in analysis with more than one hidden layer in Zeyuan Allen-Zhu, Yuanzhi Li, and Yingyu Liang. Learning and generalization in overparameterized
neural networks, going beyond two layers. NeurIPS, 2019a.

It would be really interesting to see a discussion of what is similar and different between these works and results and the present work. And what is the added value of the present one about understanding the teacher-student setting.






**Experience Assessment:**

I have published one or two papers in this area.

**Review Assessment: Checking Correctness Of Derivations And Theory:**

I assessed the sensibility of the derivations and theory.

**Review Assessment: Checking Correctness Of Experiments:**

I assessed the sensibility of the experiments.

**Review Assessment: Thoroughness In Paper Reading:**

I read the paper at least twice and used my best judgement in assessing the paper.

---

> ### Author Response · Authors · 2019-11-07
> **Reply to Review #2**
>
> The results stated in the theorems are indeed necessary conditions for any stationary density \rho with global support. The derivation of the mean field equation is formal, however, the proof from the mean field equation to the results in the theorems are not.
>
> Concerning the point of larger weights learned first, we believe that the mechanism here is
> different from the results in PCA. The phenomenon that the large eigenvalues get learned first comes as a result of power iteration and reorthogonalization, while the result here comes from the quadratic nature of the right hand side of the mean field equation. For example, the dynamics of the two equations at the end of Section 2.2 are essentially independent, while for PCA reorthogonalization is required for the smaller eigenvectors to converge.
>
> We plan to provide a non-technical summary of the findings in the revision of the paper and include the missing references for the teacher-student setting.

---

> > ### Comment · AnonReviewer2 · 2019-11-15
> > **Acknowledgement on the answer**
> >
> > Thank you for the answer. The point about PCA is interesting. In the view of the lack or a revision of the paper and of the answer being rather unspecific concerning all my other comments (except the one on PCA) I maintain my rating. Please do explain better the value and consequences of your work in the next version, I will be looking forward to read it.

---

### Official Review · AnonReviewer1 · 2019-10-24
**Official Blind Review #1**

**Rating:** 1

**Review:**

This paper studies the mean field models of learning neural networks in the teacher-student scenario. The main contributions are summarized in Theorems 2-4, which characterize the stationary distribution of gradient-descent learning
for what are commonly called committee machines. Theorem 2 is for a committee of simple perceptrons, whereas Theorem 3 is for what the authors call the ensemble-ResNet, which is in fact a committee of ResNets, and Theorem 4 is for what the authors call the wide-ResNet, which is actually a stacked committees of simple perceptrons.

These three theorems are straightforward to derive from differentiation of the expected squared loss E[\rho] with respect to \rho and equating the result with zero. The argument in Example 2 in Section 2.1 has flaws. The argument on the wide-ResNet is based on the linear approximation, which effectively reduces the stacked committee network to a large committee of simple perceptrons, so that its significance should be quite limited. Because of these reasons, I would not be able to recommend acceptance of this paper.

The authors do not seem to know the existing literature on analysis of learning committee machines. See e.g. [R1] and numerous subsequent papers that cite it.

In Example 2, the authors claim that when \sigma is an odd function \rho is a stationary distribution if and only if the difference is an even function. This claim is not true in general. Consider the case where \sigma is a sign function. Then for any function f(\theta) satisfying \int_0^\infty f(\theta) d\theta=0 (such functions are straightforward to construct), let g(\theta)=f(\theta) if \theta\ge0, and g(\theta)=-f(-\theta) if \theta<0. Then \int \sigma(x\cdot\theta)g(\theta) d\theta=0 holds, whereas g is an odd function. This is a counterexample of the authors' claim here. Many more such counterexamples are quite easily constructed on the basis of orthogonal polynomials, demonstrating that the stated stationary condition in Theorem 2 may not be so strong for characterizing the stationary distribution.

Page 2, line 30: Given a function (f -> y)
Page 2, line 38: I do not understand what the subscript $x$ of the integral sign means.
Page 2, line 40: The last term should be squared.
Page 3, line 7: Should \theta_j be \theta_j(t)? Should \rho(\theta) be \rho(t,\theta)?
Page 3: Both \rho(t,\theta) and \rho(\theta,t) appear, which are inconsistent.
Page 3, line 16: under (the the) Wasserstein metric
Page 3, line 22: \rho_0(x) should probably read \rho(0,\theta).
Page 3, line 26: "should follow normal reflection" I do not understand the reason for it. How \theta_i(t) should behave when it hits the boundary depends on how the gradient-descent algorithm is defined in such cases.
Page 6, line 7, page 7, line 18: "The GD/SGD algorithm": What is shown here is the GD algorithm and is not the SGD.
Page 7, line 14: and (vanishes -> to vanish)

[R1] David Saad and Sara A. Solla, "On-line learning in soft committee machines," Physical Review E, volume 52, number 4, pages 4225-4243, October 1995.


**Experience Assessment:**

I have read many papers in this area.

**Review Assessment: Checking Correctness Of Derivations And Theory:**

I carefully checked the derivations and theory.

**Review Assessment: Checking Correctness Of Experiments:**

N/A

**Review Assessment: Thoroughness In Paper Reading:**

I read the paper at least twice and used my best judgement in assessing the paper.

---

> ### Author Response · Authors · 2019-11-07
> **Reply to Review #1**
>
> The authors are not aware of the literature on committee machines. We plan to cite the key papers in this area in the revision. We would like to mention that the setting of the current paper is somewhat different as the student network has infinite width.
>
> Thanks for pointing out the mistake in Example 2. We will correct that in the revision. In fact, if
> \sigma is a polynomial function, one can choose rho-rho^* to be the Fourier transform of a function that vanishes in a neighborhood of the origin to get zero inner product.

---

> ### Comment · AnonReviewer1 · 2019-11-14
> **Acknowledging Rebuttals**
>
> I have read the other review comments as well as the author responses. On the basis of them, I would like to keep my initial rating, since the author responses do acknowledge my concerns but not seem to resolve them.

---

### Decision · Program_Chairs · 2019-12-19

**Decision:**

Reject

**Comment:**

This paper studies the evolution of the mean field dynamics of a two layer-fully connected and Resnet model. The focus is in a realizable or student/teacher setting where the labels are created according to a planted network. The authors study the stationary distribution of the mean-field method and use this to explain various observations. I think this is an interesting problem to study. However, the reviewers and I concur that the paper falls short in terms of clearly putting the results in the context of existing literature and demonstrating clear novel ideas. With the current writing of the paper is very difficult to surmise what is novel or new. I do agree with the authors' response that clearly they are looking at some novel aspects not studied by the previous work but this was not revised during the discussion period. Therefore, I do not think this paper is ready for publication. I suggest a substantial revision by the authors and recommend submission to future ML venues.